# National Ecosystem Services Assessment in Hungary: Framework, Process and Conceptual Questions

**Ágnes Vári** [1,*], **Eszter Tanács** [1], **Eszter Tormáné Kovács** [2], **Ágnes Kalóczkai** [1], **Ildikó Arany** [1], **Bálint Czúcz** [1], **Krisztina Bereczki** [1], **Márta Belényesi** [3], **Edina Csákvári** [4], **Márton Kiss** [1,5], **Veronika Fabók** [1], **Lívia Kisné Fodor** [6], **Péter Koncz** [1], **Róbert Lehoczki** [3], **László Pásztor** [7], **Róbert Pataki** [3], **Rita Rezneki** [1], **Zsuzsanna Szerényi** [8], **Katalin Török** [4], **Anikó Zölei** [1], **Zita Zsembery** [1] and **Anikó Kovács-Hostyánszki** [1]

1 Centre for Ecological Research, Lendület Ecosystem Services Research Group, Alkotmány út 2-4, H-2163 Vácrátót, Hungary
2 Institute for Wildlife Management and Nature Conservation, Hungarian University of Agriculture and Life Sciences, Páter Károly u. 1., H-2100 Gödöllő, Hungary
3 Lechner Knowledge Center, Bosnyák tér 5., H-1149 Budapest, Hungary
4 Centre for Ecological Research, Restoration Ecology Group, Alkotmány út 2-4, H-2163 Vácrátót, Hungary
5 Department of Climatology and Landscape Ecology, University of Szeged, Egyetem utca 2., H-6722 Szeged, Hungary
6 Department of Nature Conservation, Ministry of Agriculture, Apáczai Csere János u. 9., H-1052 Budapest, Hungary
7 Institute for Soil Sciences, Centre for Agricultural Research, H-1022 Budapest, Hungary
8 Institute of Sustainable Development, Corvinus University of Budapest, Fővám tér 8., H-1093 Budapest, Hungary
* Correspondence: vari.agnes@ecolres.hu

**Abstract:** Mapping and assessing ecosystem services (ES) projects at the national level have been implemented recently in the European Union in order to comply with the targets set out in the EU's Biodiversity Strategy for 2020 and later in the Strategy for 2030. In Hungary this work has just been accomplished in a large-scale six-year project. The Hungarian assessment was structured along the ES cascade with each level described by a set of indicators. We present the selected and quantified indicators for 12 ES. For the assessment of cascade level 4, human well-being, a set of relevant well-being dimensions were selected. The whole process was supported by several forms of involvement, interviews, consultations and workshops and in thematic working groups performing the ES quantifications, followed by building scenarios and synthesizing maps and results. Here we give an overview of the main steps and results of the assessment, discuss related conceptual issues and recommend solutions that may be of international relevance. We refine some definitions of the cascade levels and suggest theoretical extensions to the cascade model. By finding a common basis for ES assessments and especially for national ones, we can ensure better comparability of results and better adoption in decision making.

**Keywords:** mapping and assessment of ecosystem services; ecosystem services cascade; cascade framework; participation; indicators; scenarios; operationalization

## 1. Introduction

In the past few decades, as the Millennium Ecosystem Assessment (MA) has drawn attention to the rapid degradation of natural habitats and the importance of the contribution of nature to human well-being [1,2], the concept of ecosystem services (ES) has been integrated into international policies and become a central element of EU target setting and measures for nature conservation [3,4]. Action 5 of Target 2 of the Biodiversity Strategy for 2020 [5] required all member states to map and assess the ecosystem condition (EC) along with their status and economic value of the ecosystem services they provide.

The requirements also included the integration of the valuation into EU- and national-level accounting and reporting systems by 2020. An increasing amount of guidance material has been provided to member states by the European Mapping and Assessment of ES (MAES) working group only in recent years in order to help to fulfil their obligation to map and assess ES [6–9]. Nevertheless, it is still a major challenge to delineate the concept of a national MAES that is consistent and can be followed all through the assessment process, especially given the diverse aspects of a great number of nationally relevant stakeholders, the diversity of expectations to be met and the levels of complexity and aggregation to be taken into account. Since one of the main targets of the new EU Biodiversity Strategy for 2030 is the restoration and protection of ecosystems and their services [10], sharing of knowledge and streamlining based on these MAES assessments is more required than ever.

National-level mappings have already been implemented to a certain extent in many member states [11,12], which are, however, often documented in national languages and only gradually becoming available to the international community. Few states have completed their assessments and published their results (e.g., UK: [13]; Luxembourg: [14]; Spain: [15]), but in many member states, the process is not finished yet [11], and results have not been published in an easily accessible form yet. Several countries published some preliminary information, roadmaps, plans and pilots to be developed further in national assessments [16–20]. Some presented case studies are for certain areas, certain aspects and ecosystem types or for specific regions [21–23]. Even though these national MAES projects differ in many aspects, which makes it not easy to apply them in other countries, their background, conceptual considerations, methodologies and limitations are of great value for the design and planning of further assessments.

One of the major challenges of national MAES is to integrate a multitude of aspects, needs and limitations, taking into account numerous interlinkages between nature and society, ecosystem goods and services and human well-being. Conceptual frameworks can help to structure and analyze complex issues, to assist in formulating complex relationships and to integrate across disciplines and settings [24,25]. One of these is the cascade framework [26] that describes the flow of ES from nature to society along four distinct levels that received some discussions or varying interpretations of the single components (e.g., [27–31]). Broader concepts have been developed that build on the basic cascade framework are the EU MAES framework [4] or the Integrated Ecosystem Assessment Framework [32,33] with several MAES reports being published only recently [9,11]. Other similar frameworks include the System of Environmental Economic Accounting—Ecosystem Accounting (SEEA-EA) [34], developed by the UN for operationalizing natural capital assessments with several useful elements that can be also applied elsewhere (e.g., [35]), or the IPBES Nature's Contribution to People framework, emphasizing the variety of perceptions on ES in different cultures [3].

Reviewing the findings and the lessons learnt from a national MAES can add to the general ES discourse, help to refine the assessment framework and operationalize the procedure of mapping and assessment of ES. The conceptual issues presented here offer guidelines for designing a coherent workflow of ES assessment at national level but can be also useful for streamlining regional assessments. Depicting in detail the cornerstones and elements that the assessment process relies on also gives a good basis for future assessments.

A mainly EU funded program was launched in 2016 to help accomplish tasks emerging from strategies, EU Directives and international agreements, like the Biodiversity Strategy to 2020, the European Landscape Convention and others. It was established with a broad science–policy interface, with the coordinator and beneficiary being the Ministry of Agriculture, State Secretariat for Nature Conservation and numerous experts of research institutes giving their scientific knowledge to complete the assessments. It provided a unique opportunity for the cooperation of different fields of expertise and involvement of the stakeholders to support decision making. The program included four projects:

(i) the further data gathering on Natura 2000 habitats and species, (ii) the Hungarian National Mapping and Assessment of Ecosystem Services project (MAES-HU), (iii) the classification of the Hungarian landscapes based on landscape character and the (iv) assessment of the status and development of green infrastructure.

In this paper, we present the structure and process of the Hungarian National Mapping and Assessment of ES project (MAES-HU), including the most important lessons learned during its implementation, in order to collect and share the experience worth including in further assessments in the future. Accordingly, in the next sections we implement the following:

1.  Describe the process of the national mapping and assessment of ES in Hungary (MAES-HU) from ecosystem type mapping, the selection of relevant ES and their indicators at the cascade levels to their mapping;
2.  Discuss the methodology and conceptual considerations in MAES-HU, in particular on the following:
    a.  Mapping ecosystem types and their interactions;
    b.  Choosing indicators for ecosystem condition, ecosystem services capacity and actual use;
    c.  Relating them to the cascade framework, suggesting some extensions;
    d.  Relating them to aspects of human well-being.

We describe first the process of the Hungarian MAES, and then discuss conceptual insights that result from the assessment and the methodological considerations taken. We relate these to applications and findings in similar national-level projects.

## 2. Methods and Process of the Hungarian Mapping and Assessment of ES

The national Mapping and Assessment of ES project (MAES-HU) was implemented between 2017 and 2022, with a preparatory phase in 2016–2017, and aimed at mapping and evaluating a set of prioritized ES, along with ecosystem extent and condition. The mappings and assessments had to rely on existing databases, as the project did not include the primary collection of new data. The base year of the analyses was set to be 2015. The project laid special emphasis on producing a detailed ecosystem type (ET) map [36] and on assessing and mapping a set of partly pre-defined ecosystem condition (EC) indicators for the whole country [35]. The conceptual framework of the ES assessment was provided by the cascade model [26] according to which MAES-HU set out to evaluate the selected ES at all four levels, showing the flow of ES from nature to society: (1) ecosystem condition, (2) ES capacity (= potential), (3) actual use (= flow) of ES and (4) ES contribution to human well-being as the scheme shows in Figure 1. Economic valuation was also carried out to complement the assessments for selected ES. The assessment of ES along the cascade was complemented by a scenario planning exercise and a synthesis of the results. Involvement of stakeholders and experts in the whole procedure was an important element. Figure 2 shows the sequence of tasks in MAES-HU. In the next chapters we give a detailed overview of these elements.

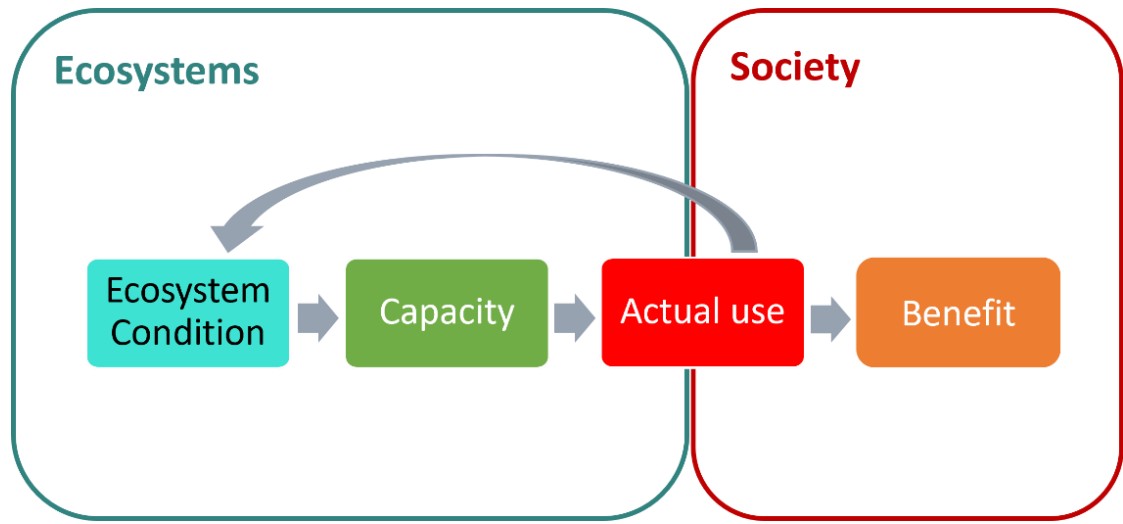

**Figure 1.** The basic ecosystem services cascade (adapted from [26]) with the ecosystem condition (i.e., ecosystem features, characteristics, properties, structures), ecosystems' capacity (or: potential) to deliver ecosystem services (i.e., functions), the actual use (or: flow) of ecosystem services (i.e., that part of the capacity that is used by humans), benefits (in terms of any increase in human well-being) and society's effect on ecosystems and their condition (pressures).

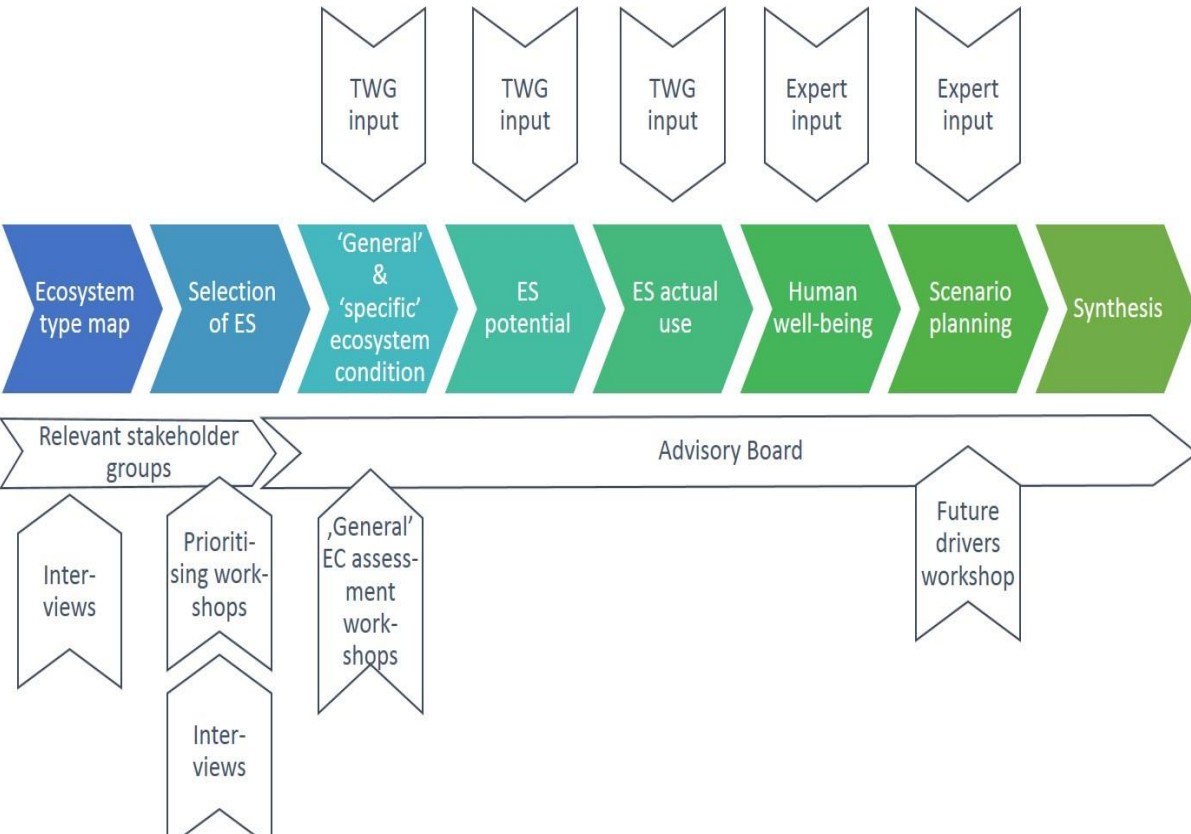

**Figure 2.** The workflow of MAES-HU showing the major steps of the process from establishing the ecosystem type map to synthesizing the results (main line of colored arrows) marking the points at which some participation took place (in the form of interviews, workshops with external experts, Technical Working Group (TWG) workshops or individual expert consultations), (uncolored arrows).

### 2.1. Participation

The involvement of different stakeholder and expert groups was regarded as important right from the beginning of the project (see also Figure 2). Here we present the main steps of involvement. Stakeholder involvement already started at the preparatory phase and continued during the whole project. A series of interviews (22 in total) was conducted with representatives of the most important sectors (nature conservation, forestry and hunting, agricultural, angling/fishing, water management, spatial planning, transport and infrastructure, tourism and industry) in order to get acquainted with the sector representatives' responsibilities, their interests and activities in relation to ecosystems and ecosystem services. The selection of ES was assisted by a series of prioritization workshops where the representatives of these sectors participated (see Section 2.3.1). The selected ES were divided into six thematic groups for mapping and assessment in the project. Six 'Technical Working Groups' (TWGs, consisting of 82 experts altogether) related to the thematic groups of ES were set up that suggested methods, available data and the process of how to assess the selected ecosystem services and then performed the assessments. Another expert group was also formed later on to give conceptual advice on the assessment of human well-being. Further experts were involved to assist in scenario building. Workshops were also held to support the assessment of the 'general' EC indicators and evaluate possible land-use scenarios. The whole project was accompanied by an Advisory Board (29 members) of sectoral leaders, decision makers and a few scientists that worked as a consulting and legitimating body. Other forms of involvement were applied throughout the whole project (e.g., personal consultations). In total, 236 experts, stakeholders or decision makers participated in the project.

### 2.2. Ecosystem Type Mapping

For many national or large-scale ES assessments in Europe, Corine Land Cover (CLC) [37] is used as the main resource underlying their ET map. The popularity of CLC is understandable, because it covers the entire Pan-European region with relatively rich spatial and thematic detail, and it is freely available. However, initial expert discussions in MAES-HU revealed that the ecological specificities of the country are not covered adequately by CLC, and there was a great need for a thematically and spatially more refined new national ecosystem type (ET) map covering the whole country. This ecosystem type map was the basis of EC and ES mapping. The map uses a three-level hierarchical classification: the six broad ecosystem types at the first level (urban areas, agricultural lands, grasslands, forests, wetlands and waterbodies; based on [7]) were further refined, with fine thematic detail represented in 56 third-level classes. These reflect mainly land cover and to a lesser extent land-use type. The map was created in the form of a raster, with a spatial resolution of 20 × 20 m. The reference year was 2015, but some databases available only for 2016 and 2017 were used as well. The ET map was created with an iterative combination of sectoral databases, such as the National Forestry Database [38]; the Land Parcel Identification Scheme [39]; other national thematic GIS layers (detailed habitat maps); the Digital, Optimized, Soil-Related Maps and Information in Hungary (DOSoReMI.hu; [40]); and international databases, such as the Copernicus High-Resolution Layer (HRL), Water and Wetness (WAW) [41] and certain elements of OpenStreetMap. Data gaps were filled using image-based predictive mapping [42], integrating remote sensing (Sentinel-1 and -2) and environmental data (soil information and topographic indices from a national DEM) using a Random Forest [43] classifier. The integration of the different databases in a theoretical data cube [44,45] is an example of the application of a relatively new concept in the use of geospatial data. The map and its creation process are described in detail in [36]. The National ET map (Ecosystem Map of Hungary) was completed in 2019 and is freely available for viewing and downloading [http://alapterkep.termeszetem.hu/] (accessed on 1 June 2022.

### 2.3. Assessing Ecosystem Services along the Cascade

2.3.1. Selection of Ecosystem Services for Assessment

As the first step towards the selection of ES, the Common International Classification of Ecosystem Services (CICES 4.3, [46]) was translated and adapted to the Hungarian context including all categories from the original system with the exception of marine ES irrelevant for Hungary (see CICES-HU in). The selection of ES was based on the sectorial leader interviews. Altogether, 73 ES items were mentioned by the leaders in the interviews that were categorized as a next step based on CICES-HU and then prioritized in four participatory workshops (with altogether 42 participants) according to the following criteria:

1. Number of nominations as 'most important' in the workshops;
3. Emphasis on regulating services and relevance for nature conservation;
4. Relevance for society and for decision makers;
5. Easy to communicate;
6. Availability of relevant data.

These were also raised in [17,47,48] and in the requirements by the project to aid nature conservation in the long term.

ES prioritization was performed for six broad ecosystem types delineated in compliance with [7], similarly to the six broad ET of the Ecosystem Map: with both water and wetland united in one group, and urban and arable lands united in one group; the six types are forests, water bodies and wetlands, settlements, grasslands and arable lands. Eventually, 8–10 ES per broad ecosystem type were selected by consensus and synthesized into a set of 12 ES (CICES classes), split into 16 ES items to be assessed (Table 1). In contrast to many other national MAES, ES were primarily organized thematically and not according to ecosystem types. The selected ES were assessed by six technical working groups (TWGs), each one focusing on one or a few related ES (Table 1). Some of the selected ES items were split in order to reflect the biophysical processes and treated by more than one TWG. Thus, microclimate regulation was dealt with at the landscape scale by the Climate and Energy TWG and at the settlement scale by the Urban TWG. Filtration processes within water and soil were analyzed by the Hydrology TWG, while processes concerning air pollution were dealt with by the Urban TWG. In general, all assessments aimed at ES mapping at the national level with whole country coverage. Exceptions were the ES assessed within the Urban TWG, which focused on four sample cities and suburbs.

**Table 1.** Assessed ES items in MAES-HU, the Technical Working Groups (TWG) handling them and their correspondence to CICES 4.3 classes. Some ES are split between several TWG due to scale reasons. TWGs' abbreviations: FOOD—Food production; CLIM—Climate and Energy; HYDR—Hydrology; URB—Urban; POLL—Pollination; CULT—Cultural.

| ES Name in MAES-HU | MAES-HU Short Name | ES Definition for MAES-HU | TWG | | CICES 4.3 Classes |
|---|---|---|---|---|---|
| Cultivated crops for nutrition | Cultivated crops | cultivated crops (major arable crops *, fruits, vegetables and vines) and hay for nutrition | FOOD | 1.1.1.1 | Cultivated crops for nutrition |
| Reared animals and their products | Reared animals | reared animals and their products used for nutrition | FOOD | 1.1.1.2 | Reared animals for nutrition |
| Firewood | Firewood | timber used for fuel | CLIM | 1.3.1.1 | Plant-based resources for energy |
| Filtration of water soluble pollutants | Filtration of water | filtration of diffuse pollutants (phosphorous) from agricultural effluents | HYDR | 2.1.2.1 | Filtration/sequestration /storage/accumulation by ecosystems |
| Filtration of air pollutants (urban) | Filtration of air | deposition of particle pollutants in settlements | URB | 2.1.2.1 | Filtration/sequestration /storage/accumulation by ecosystems |
| Control of soil erosion | Erosion control | protection against water erosion by natural or planted vegetation | HYDR | 2.2.1.1 | Mass stabilisation and control of erosion rates |
| Flood regulation by water retention | Flood regulation | (rain)water retention and buffering by vegetation on slopes | HYDR | 2.2.2.2 | Flood control |
| Drought mitigation | Drought mitigation | mitigation of droughts by water storage in the landscape | HYDR | 2.2.2.1 | Hydrological cycle and water flow maintenance |
| Flood regulation in floodplains | Flood regulation in floodplains | flood risk mitigation and buffering by floodplains | HYDR | 2.2.2.2 | Flood control |
| Management of rainwater (urban) | Urban flood regulation | (rain)water retention and buffering by vegetation in settlements | URB | 2.2.2.2 | Flood control |
| Pollination | Pollination | pollination by wild bees | POLL | 2.3.1.1 | Pollination and seed dispersal |
| Global climate regulation | Global climate regulation | global climate regulation by reducing the amount of greenhouse gases | CLIM | 2.3.5.1 | Global climate regulation by reduction of greenhouse gas concentrations |
| Microclimate regulation at landscape level | Regional microclimate regulation | regional climate regulation at landscape level outside settlements | CLIM | 2.3.5.2 | Micro and regional climate regulation |
| Microclimate regulation (urban) | Microclimate regulation | mitigation of summer heat stress in settlements | URB | 2.3.5.2 | Micro and regional climate regulation |
| Recreational use of nature | Recreation | recreational use of nature by hiking | CULT | 3.1.1.2 | Use of nature for recreation |
| Cultural heritage | Cultural heritage | aggregation of activities, knowledge, norms and elements of identity related to mushroom picking | CULT | 3.1.2.3 | Cultural heritage |

* major crops: wheat, corn, rapeseed, barley and sunflower.

2.3.2. Ecosystem Condition Indicators in MAES-HU

The assessment of ecosystem condition in MAES-HU was conducted in two distinct ways in different parts of the project. In one part, we designed 'general' condition indicators aiming to describe the level of human impact on ecosystems. This approach is closely related to the earlier concepts of ecosystem integrity and ecosystem health [49,50]. Such general EC indicators were chosen that underlie several ecosystems and ensure their functioning: soil fertility, naturalness and habitat diversity. These were key aspects, evaluated with different approaches within the six broad ecosystem types together with experts on the six ecosystem types (25 people) over the whole landscape where relevant/possible (for details see [35]). The EC maps had to cover the entire area of the country, but for most ecosystem types, availability of data or availability of good quality data was an issue. Therefore, for data-scarce ecosystem types (such as grasslands and wetlands) these complex EC indicators were compiled mostly using proxy indicators of anthropogenic pressure [35].

In addition, we also chose 'ES-specific' condition indicators to be included in the cascade model for certain ES or thematic groups of ES at the first cascade level. 'ES-specific' indicators are those 'underpinning' features of an ecosystem or landscape that provide the basis for the production of a specific ES or have a well-documented influence on the provisioning of that ES, such as soil fertility for crop production or the share of green spaces and water surfaces for urban microclimate regulation (see also Table 2). Not many clear relationships are known between different EC aspects and their influence on the delivery of specific ES, but where knowledge was available, we included 'ES-specific' EC indicators in our assessments. These were developed and evaluated by the six TWGs. For many ES the respective indicators at level 1 (EC) were based on existing data (like soil fertility, soil erodibility) or on the 'general EC' indicators developed within the project (e.g., naturalness, share of green spaces and landscape heterogeneity).

**Table 2.** Compilation of the indicators selected in MAES-HU along the ecosystem services cascade with 'specific' condition indicators relevant for the specific ES at cascade level 1 and cascade levels 2 and 3 (capacity and actual use).

| | Indicators for Each ES at Each Cascade Level (Name [Unit]) | | |
|---|---|---|---|
| MAES-HU Short Name | Cascade Level 1 | Cascade Level 2 | Cascade Level 3 |
| Cultivated crops | soil fertility (relative scale) | potential yield (crops, hay); maximum long-term yield (fruits, vegetables) (t/ha) | actual yield (crops) (t/ha) |
| Reared animals | soil fertility (relative scale) | yield (livestock; products) based on potential fodder production (from crops) (t/ha) | actual yield (livestock units); actual production (meat, milk, eggs) (aggregated numbers) |
| Firewood | forestry stocks [$m^3$/ha]; naturalness of forests | mean annual increment ($m^3$/ha/y) | harvested amount of firewood ($m^3$/ha/y) |
| Filtration of water | soil hydrologic capacity; for water bodies: biotic water quality (relative scale) | relative filtering capacity of the ecosystem (relative scale) | - |
| Filtration of air | share of green spaces and water surfaces | leaf area index (plant surface available for deposition and filtration) | amount of air pollutants removed (g/$m^2$) |
| Erosion control | soil erodibility (t × ha × h × ha$^{-1}$ × MJ-1 × mm$^{-1}$) | prevented soil erosion in an optimal ecosystem state (t/ha/year) | prevented soil erosion in the actual state of ecosystems (t/ha/year) |
| Flood regulation | soil hydrologic capacity (relative scale) | relative water-retention capacity of the ecosystem (relative scale) | amount of precipitation retained by the ecosystem (mm/y) |
| Drought mitigation | soil water storage capacity (in top 2 m) [mm/2 m] | potential areas of water storage (inland-water prone areas) | - |
| Flood regulation in floodplains | 100-year return flood areas compared to actually floodable area | artificial storage areas (flood retention basins) ($m^3$) | demand: 1000 year return flood areas |
| Urban flood regulation | share of green spaces and water surfaces | runoff retention potential of the vegetation (CN parameter and Leaf Area Index) (relative scale) | amount of water intercepted on leaves ($m^3$/ha) |
| Pollination | amount of foraging resources (flowers) and nesting suitability for wild bees | relative pollination potential of wild bees in an area (relative scale) | relation between demand (insect pollination need of different crops) to pollination potential; non-cropland: flower availability (relative scale) |
| Global climate regulation | carbon stocks (above and below ground biomass, soil C) (tC/ha) | gross greenhouse gas balance (without losses, e.g., due to timber harvesting) (t $CO_2$ eqv/ha/y) | net greenhouse gas balance: recorded maximum of net C sequestration; recorded minimum of greenhouse gas emission (t $CO_2$ eqv/ha/y) |
| (Regional) microclimate regulation | local climate index (relative scale) | potential evapotranspiration (mm/y) | effective precipitation (mm/y) |
| Microclimate regulation | share of green spaces and water surfaces | evapotranspiration coefficient + Leaf Area Index (LAI) (relative scale) | bioclimatic index |
| Recreation | natural attractions: naturalness index, protection status, water proximity and landscape heterogeneity (relative scale) | natural and built attractions: hiking trails, (built) points of interest and accessibility (relative scale) | touristic use intensity: visitor and guest numbers for sample areas in national parks and accommodations |
| Cultural heritage | site suitablility for mushrooms (relative scale) | potential mushroom picking areas | actual sites for picking mushrooms; number of people visiting the sites |

2.3.3. Assessing Ecosystem Service Capacity and Actual Use

For the 12 selected ES items the TWGs elaborated indicators and assessments based on available data for cascade levels 2 and 3 (potential ES = ES capacity and flow = actual use of ES; Table 2). For this, a hierarchical approach was followed, first targeting the development of simple (tier 1) matrix models [51,52] to map ES capacities (cascade level 2). Such models, with the ET map being their only spatial input, are no more than simple

'lookup tables' that link the ecosystem types to indicator scores and can then further be refined by rules including more specific spatial information (tier 2 models). If expert knowledge and data were available, TWGs developed tier 3 (e.g., process-based) models for certain ES (Table 3).

Some of the selected indicators were general and easy to use, like specific metrics of landscape features (e.g., share of green spaces, proximity of water and track density), while some others were very specific, and for their interpretation and application, expert knowledge is needed (e.g., CN parameter ('curve number') or water retention capacity). The applied models at level 2 of the cascade (ES capacity models) ranged from tier 1 to tier 3 models, including highly complex biophysical models (based on the model Biome-BGC-AgroMo [53], for carbon sequestration/global climate regulation and potential crop production), with the majority relying on rule-based models of intermediate complexity (e.g., adoption of the ESTIMAP model for pollination; [54]). Some models also included data derived from remote sensing (such as Leaf Area Index for filtration of air, microclimate regulation or flood regulation). For cascade level 3, the actual use of ES, indicators were mostly either modelled (e.g., global climate regulation), existing statistical data were used (e.g., crop production) or an alternative assessment of their "importance" was chosen by assessing the demand for the service (pollination, flood regulation or recreation). It was only for the three provisioning ES crop production, reared animals and firewood that relevant statistical data were available, and thus were used to represent the actual use of ES (see Table 2).

**Table 3.** The types of input data and models used for assessing the selected ES in MAES-HU with different levels of complexity: tier 1: solely expert judgement; tier 2: rule-based matrix models; tier 3: biophysical, hydrological or meteorological models. *—only for case study areas. Detailed reports on methods and results available at http://termeszetem.hu/hu/okoszisztema-szolgaltatasok/tanul-manyok-szmcs (accessed on 1 June 2022).

| MAES-HU Short Name | Level 1—Type of Input | Level 2—Model Type | Level 3—Assessment Type |
|---|---|---|---|
| Cultivated crops | existing data (national soil database) | biophysical model for crop and grassland (Biome-BGC); long-term statistical data for fruit and vegetable | statistical data—national databases on crop production; expert judgement on hay production |
| Reared animals | existing data (national soil database) | expert judgement (grasslands) + calculated from crop biophysical model (cropland) | statistical data—national databases on animal production |
| Firewood | existing data (national forestry database); general EC indicator | expert judgement based on timber harvesting tables from national forestry database | statistical data—national databases and national survey on use ratio of harvested timber |
| Filtration of water | biophysical model | rule-based matrix model (expert judgement) + biophysical model for EC; for water: existing data (components of the Water Framework Directive monitoring) | * InVEST, SWAT |
| Filtration of air | general EC | existing data (remote sensing) | * modelled (EMEP-MSC-W model) |
| Erosion control | existing data | rule-based (empirical) matrix model with expert judgement for vegetation factor (USLE based) | rule-based (empirical) matrix model with expert judgement for vegetation factor (USLE based) |
| Flood regulation | biophysical model | rule-based matrix model (expert judgement) + biophysical model for EC | * InVEST, SWAT |
| Drought mitigation | hydrological model | existing model with expert judgement | - |
| Flood regulation in floodplains | existing data | existing data | demand: existing data/map |
| Urban flood regulation | general EC | basic matrix (expert assessment); existing data (remote sensing) | * modelled (i-Tree) |
| Pollination | expert judgement | rule-based matrix model (expert judgement based on ESTIMAP) | literature data and statistical data on crops; rule-based matrix model |
| Global climate regulation | existing data (for soils, forests); biogeophysical models (for grass— & croplands) | biophysical model (Biome-BGC); rule-based model (IPCC method—National Greenhouse Gas Inventory (NGHGI)) | biophysical model (Biome-BGC); rule-based model (IPCC method—National Greenhouse Gas Inventory (NGHGI)) |
| (Regional) microclimate regulation | expert judgement | biophysical (meteorological) model | meteorological model based on soil data |
| Microclimate regulation | general EC | basic matrix (expert assessment); existing data (remote sensing) | * modelled (InVEST Urban) |
| Recreation | ESTIMAP-based expert evaluation | rule-based matrix model (expert judgement based on ESTIMAP) | * statistical data for case study |
| Cultural heritage | rule-based matrix model (expert judgement biophysical model) | rule-based matrix model (expert judgement + questionnaire data); biophysical modelling | questionnaire data |

* only for case study areas.

2.3.4. Assessing Human Well-Being

Within the project a qualitative assessment of the contribution of ES to human well-being was conducted with the participation of the Technical Working Groups. Six main components of well-being were chosen based on related scientific literature:

1. Material welfare;
7. Health;
8. Environmental security;
9. Community and social relations;

10. Self-fulfillment and self-esteem;
11. Participation, freedom of decision and action.

An attempt was made to define the links between each ES and each component of well-being except for participation, freedom of decision and action, which was considered to be determined more by social and institutional setting. For each link the main stakeholder groups were also identified. The links were evaluated on a 1–3 Likert scale (1—not relevant/not important; 2—slightly important; 3—very important).

The well-being working group that was set up at a later stage of the project proposed a modification on the components of well-being, reducing them to three components:

1. Health including physical, mental and social health and self-fulfillment;
12. Environmental security;
13. Participation, freedom of decision and action.

Material welfare was seen by the experts of the working group as a means for well-being; therefore, it was suggested not to be included as a separate component of well-being. Community and social relations as well as self-fulfillment and self-esteem were suggested to be merged into the health component.

Besides the qualitative assessment of the links between well-being components and ES, economic valuation of three ES was conducted in MAES-HU: carbon sequestration, flood regulation and recreation (specifically hiking). After an extensive literature review, the following methods were applied: avoided costs (for carbon sequestration and flood regulation), replacement costs (for flood regulation) and benefit transfer (for recreation). For the valuation available data were used. It largely built upon the results of the ES assessments prepared by the respective expert groups in the project.

*2.4. Scenario Building*

Exploring possible future scenarios makes it possible to go beyond the limitations of present-day land use, or of the single cascade levels as defined. The scenarios created in the framework of MAES-HU primarily reflect on the natural environment, the ecological condition of Hungary and the future capacity of the ecosystems to supply services. Their aim was to draw attention to future opportunities, uncertainties and threats. The process consisted of three main parts: (1) identification of the main drivers influencing the present and future condition of ecosystems and their services; (2) formulation of the scenarios (scenario building); and (3) their assessment in terms of future land use, land cover and ES. Steps (1) and (2) were performed by experts in an online Delphi process, while step (3) also involved a targeted workshop and is being continued.

Drivers influencing ecosystems can be direct (such as climate change, the emergence of invasive alien species or land-use change) and indirect (such as economic, technological or demographic changes), the latter determining the direction of the current and expected development of society and thus influencing the direct factors. The identification of drivers provided the basis for scenario building, and it was done by a group of experts, exploring (1) the direct and indirect factors that are most influencing the present and future of Hungary's ecosystems, (2) correlations between direct drivers and the state of the natural environment and (3) causal relationships between indirect and direct drivers.

The scenarios were based on two internationally renowned scenario-building processes, one from the UK National Ecosystem Assessment and the other from the IPBES—Regional Assessment Report on Biodiversity and Ecosystem Services for Europe and Central Asia [13,55]. Five scenario archetypes were identified and adapted to the Hungarian environment at the national level (see Table 4).

**Table 4.** Future scenarios developed in MAES-HU (based on UK NEA and IPBES scenarios), with their main features.

| Scenarios | Main Tendencies |
|---|---|
| Business as usual | Current economic, social and technological trends continue unchanged (reference scenario). |
| The market solves everything | Scenario based on economic growth and technological solution to environmental problems. |
| National sovereignty | Due to the growing disparities in economic development, global development trends based on international cooperation are coming to a halt. The world is falling apart into independent regions, among which mistrust is growing. |
| Self-determination of local communities | Society's awareness is growing towards environmental and social sustainability at the regional level around the world. |
| Centralized sustainability | Both the public and leaders show a proactive attitude towards environmental problems, which are addressed through global cooperation and strong regulation. |

The final step of the process is the evaluation and quantification of scenarios. In doing so, we first provided probabilistic estimates of how the proportion of major ET would shift for each scenario, including ET changes arising from radical land-use changes like reforestations, land abandonment or floodplain reconnections. Resulting changes in the availability of the selected ES follow. The process of scenario quantification, especially its spatial definition, has not been completed yet; therefore, results are available as summed for the whole country. The Decision Support Systems tool (DSS, developed on the Geonamica software platform, Research Institute for Knowledge System—RIKS), which is also used in the Hungarian spatial planning process, is able to model the complex interactions between socio-environmental processes by identifying complex links between drivers and land-use categories and interpreting the scenarios on alternative ET maps. The follow-up project of MAES-HU plans to adapt the DSS tool for national-level quantification and mapping as well as for pilot areas with more detailed scenario evaluation and specific models.

*2.5. Integration and Synthesis*

Integration between the results of different assessments in terms of analyzing and synthesizing within one system is a final step of a complex ES assessment. In contrast to a quick and less in-depth analysis where one big matrix is filled in at one or few workshops (as in [24,51,52,56]), the analyses of different working groups and different models with different interpretations of indicators need to be aligned in a dedicated step.

The assessments focusing on the selected, individual ES were synthesized, ES bundles typical for certain regions were delineated, and including some EC indicators, synergies and trade-offs were revealed. Correlations, networks and hierarchical cluster analysis were used for calculating statistical relationships and visualizing links between ES and some EC indicators. Multifunctionality indices calculated from the whole set of available ES at a site were used for delineating hotspots of ES delivery within the country and to make comparisons between and within ecosystem types in this respect (e.g., how many ES are provided by forests compared to agricultural land or how many ES are provided by beech forests compared to pine plantations). The analysis was carried out at the capacity level, as this level showed the greatest coverage across the assessments of the 12 ES [57].

**3. Discussion of Conceptual Questions and Insights**

In the following sections, we critically analyze the process of the national mapping in Hungary, the decisions taken, their theoretical background and their practical implications. The topics that we reflect upon cover several conceptual issues mainly, but not

solely, regarding the interpretation of the ES cascade levels and the cascade model's application.

### 3.1. Participation—Who to Involve and Why?

To ensure broad-scale scientific, policy and social credibility, active involvement of stakeholders and experts was of utmost importance during the whole assessment process. Participation helps to anchor assessments and assure scientific credibility as well as enhance policy uptake [32,58]. In MAES-HU, it also helped to bridge data and information gaps, identify and access the data needed, develop new methods for assessments and combine relevant bits of knowledge and experience.

Selection of ES is often completed as desk research, as it is strongly dependent on data availability (e.g., [19,22,59]) and does not include any participatory elements (but see [18] for Slovakia). A review of several European national-level ES processes [12] showed that the most common stakeholder groups identified and considered were ministries, environmental administration and academic institutions in all cases, as well as NGOs and private sector institutions in most cases. However, in most cases, they were involved only in defining user needs and some initial views, and only in a few cases were they involved in the actual assessment or scenario building. The study by the authors of [32] identified unsatisfactory stakeholder involvement as a gap in European MAES processes. In MAES-HU, sectoral leaders and the respective ministerial organizations were considered as the main stakeholders, representing their specific needs and interests during the assessment. Their involvement was ensured through interviews, workshops and the advisory board. On the other hand, experts holding specific, scientific knowledge related to different thematic parts of the assessment were also seen as important. They were involved as members of the technical expert groups but also as participants in workshops or through personal communication.

While expert and stakeholder involvement in the process was well established, inclusion of the wider public was not an objective in the Hungarian assessment as this was beyond the project's capacity. The involvement of the broader public seems to not usually take place in other national-level assessments either [12].

### 3.2. Ecosystem Types—How Can we Represent Their Interactions?

The results and accuracy of ES assessments are highly scale dependent [60,61]. As the ET map formed the basis of the assessments in MAES-HU, its spatial resolution of 20 m defined the lowest possible scale of the mapping and assessments. The cell size is lower than the size of most habitat patches characterizing the Hungarian landscape; thus, mixed pixels are relatively rare. This has an effect on the applicability of certain indicators, such as, e.g., ET ratios, which can be not applied at the cell level. However, this allows for landscape-level ratios to be calculated in a more precise way.

Ecosystem mappings within Europe often rely on Corine Land Cover and are thus of much coarser spatial resolution [20,22,62]. As the less detailed spatial and thematic resolution limit their usability for management and planning, many of the member states have recently created new ecosystem maps to serve as a basis for ES assessments (e.g., [63–65]). See [36] for a more detailed discussion of international approaches to ecosystem mapping. Interactions between specific ETs cannot be reflected by the matrix-type ES models due to the nature of this approach to link ES to distinct ETs. A mosaic of ETs or edges between different ETs (i.e., ecotones) can be of enhanced importance for several ecosystem functions, resulting in synergistic effects of valuable ETs, meaning that their co-occurrence or spatial combination is more than their "sum" (e.g., [66,67]). In MAES-HU, the landscape-level handling of ETs was accomplished in a more general way: landscape-level features like "landscape diversity" and "ratio of semi-natural areas" were calculated in relation to each other and handled as indicators of general EC [35]. This is also in line with the Ecosystem Condition Typology of the SEEA defining a separate category for "landscape

level" EC aspects [68]. This "general" type of EC indicator was also relevant and integrated into the models for certain ES (see recreation and pollination).

In the synthesis part of the work, the relationships between the different ES and between ES and EC were studied at the national and at landscape scale (separating the mountainous areas from the lowlands) but also within the individual ETs (for example within forests or grasslands separately; [57]). Some ES appear heavily clustered in the landscape, but this may be due to their strong association with certain types of land cover (e.g., timber yield in forests; [69]). Therefore, we considered it important to differentiate between patterns that arise due to the presence of different ETs in the landscape and patterns within major ETs.

### 3.3. Application of the Cascade Model

The strength of the cascade model with its four levels of ES flow from nature to society lies in the integration of the different aspects that the different levels represent and which are often characterized by indicators from different disciplines, presenting biophysical, social and economic parameters [26,70,71]. As the authors of [7] suggest, national MAES studies should also follow the cascade framework. Most assessments make use of the cascade concept and use the ideas set forth in the concept to frame their work, but only few actually include an assessment of more than one level, even at the regional level (e.g., [24,72,73]) targeting, for example, "supply-demand" (mis)matches (i.e., the relationship between capacity and actual use). There are very few published examples where a complete mapping and assessment at all cascade levels are produced (see discussion in [30] and [74]) and even less at the national level—MAES-HU is among the first ones to have attempted this.

While it is a challenge to design an assessment that encompasses all levels, we believe that this is the best option to deliver the message of the original ES concept and make the public or decision makers aware of the vital dependence of humans on well-functioning and healthy ecosystems and the linkages between ecosystem condition and human well-being.

If we want to actually carry out an assessment and create maps, we need a set of indicators that can be quantified and measured. Thus, not only theoretical indicator development (as in [75]), but also the availability of appropriate data is of great importance, as well as the quantification (or: the valuation algorithm) of the selected indicators (see also [25,35]). During several steps of the assessment, we encountered difficulties of how the chosen terms and the chosen indicators should be filled with data-based (and model-based) content.

### 3.3.1. What Are Relevant, Specific Ecosystem Condition Indicators?

In contrast to a widespread view [32], we chose not to regard abiotic components in themselves as reflecting EC but as biophysical background variables (e.g., slope, rainfall, climatic components, or physical soil properties; see Figure 3). While these can all be important in modelling ES capacity, they are (relatively) stable topographic or climatic components defined by the location of the given pixel/spot and less sensitive to short-term changes in the integrity of the ecosystem. This complies with the criteria set out by now in the SEEA-EA for Ecosystem Condition Characteristics [34,68].

In MAES-HU, we did not define the first level of the cascade merely as the basic structure of the ecosystems (as in earlier works, e.g., [26,75]), but we also added a normative aspect to our selection criteria, so that the selected indicators would be able to distinguish between "good" and "bad" conditions [76]. In addition, we also aimed to catch those features of the ecosystems that describe their integrity connected to the delivery of ES in a mechanistic way (as set out in [50,76]) and that can be therefore linked to ES models [77]. This could be implemented more in the ES-specific condition indicators but were less required for general EC indicators (see also [35]). Some good examples for normative EC indicators—i.e., indicators with a clear directionality (the higher the better) that depict

specific features of ecosystem integrity were 'soil hydrologic capacity', which could be linked to two water flow-related ES (flood regulation and filtration of water); 'soil fertility' for crop production and reared animals (via feed production); or 'share of green spaces', which is relevant for many urban ES.

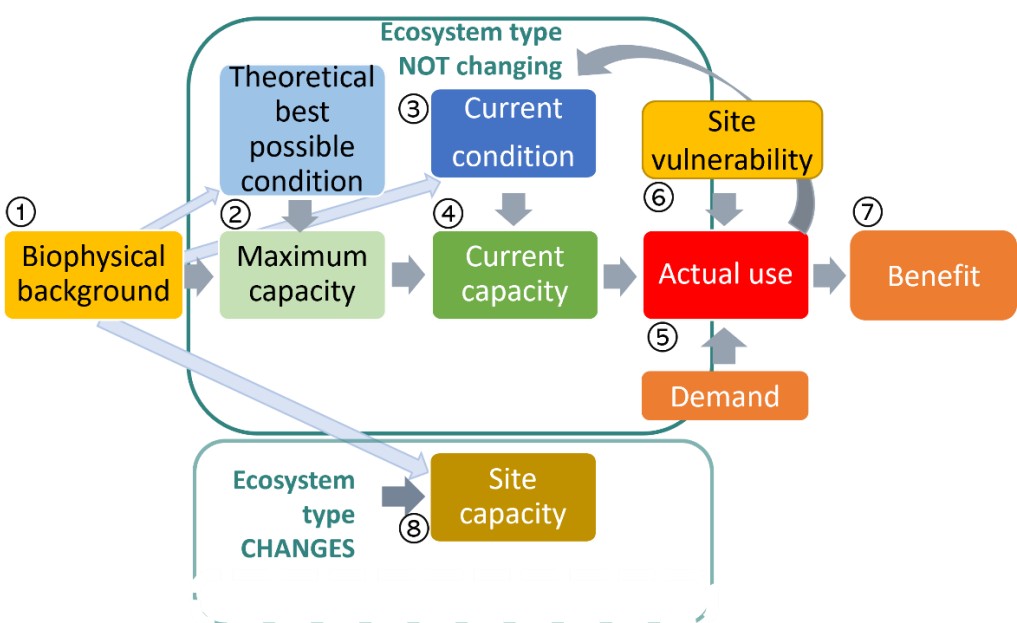

**Figure 3.** An extended cascade as suggested in MAES-HU with fine-tuned levels: (1) biophysical background as separate from ecosystem condition, (2) maximum ES capacity that is achieved only under optimal ecosystem conditions, (3) the current condition of the ecosystem and (4) the current capacity to deliver ES. (5) The actual use of ES depends on current capacity as well as on the demand from society and can be also influenced by (6) site-specific relevance or site vulnerability. (7) This results in an increase in benefit/human well-being. For all these items, the ecosystem type does not change. If it does, (8) a site capacity can be assessed for a specific ES in a different ecosystem (in terms of what ES could another ecosystem type provide; see also Section 2.4).

### 3.3.2. How Can We Define Ecosystem Service Capacity Operationally?

The delineations of the cascade levels need to be further clarified, in order to avoid overlaps, double-counting and thus confusions within (one or several) assessments [25,28]. Defining clear categories enables us to compare results and analyze them in a common frame [78]. Specifically, the delineation of ES capacity is needed for performing an analysis of the interactions, the potential synergies or trade-offs at this level. In MAES-HU, this was of specific importance, as the different ES were assessed in parallell by separate working groups. Many assessments (e.g., [27,79,80]) define ES capacity similar to the description in the ESMERALDA Glossary as "*the natural contributions to ES generation (…corresponding to) the amount of ES that can be provided or used in a sustainable way in a certain region (...over) a sufficiently long time period.*" [81]. However, this definition does not provide a clear basis for delineating the timeframe and the condition in which the ecosystem provides that ES and is not explicit about the ecosystem regarded.

We defined ES capacity (= the capacity, or the potential of an ecosystem to deliver ES) in a similar way, as the ES that the given ET in its current condition—including the present land use—is able to provide. This has some implications for the assessment that are worth emphasizing:

1.  The only ET assessed is that which is present at the given time. The ES capacity of other vegetation types possible at that location (e.g., potential natural vegetation (PNV)), or ET changes related to major changes in land use (e.g., forest clearing for agricultural areas or habitat restoration) are not considered;

14. No changes in geomorphology or hydrology (e.g., removal of dams and weirs, open-cast mining) are considered, and only the present surface and present water flow regime are considered;

15. A change in condition affects the capacity of the ecosystem to provide this service; see Figure 3 for more detail.

By adopting this definition, we deliberately refrained from assessing an ecosystem service capacity resulting from major changes in vegetation (i.e., across broad ET classes) or in any other major influencing biophysical feature (e.g., topography/geomorphology). Thus, evaluations of "potential floodplains" in areas protected by dams, or "potential carbon sequestration of forests" in non-forested areas were not part of the ES capacity assessment, but rather that of future scenario building (see Sections 2.4 and 3.4). This is also in line with the guidelines along which PNV is assessed [82,83], the definition given by [27] and that used in the SEEA accounting framework [34]. In contrast, minor changes in vegetation (remaining within the broad ET) can be seen as changes in their condition, caused by (minor) changes in land-use practice (e.g., changes in mowing frequency resulting in differing species composition). Degradation (or amelioration) of the current condition of an ecosystem decreases (or increases) the current capacity to deliver that ES, in line with the definition given in [27]. Only under "theoretical best conditions" can the theoretical greatest capacity ("maximum capacity") be achieved at that site (i.e., with the given biophysical parameters); Figure 2 visualizes these items and their relations. Without framing it in terms of theoretical/actual condition and capacity, a similar approach is taken by a Bulgarian case study in the Central Balkans [84].

### 3.3.3. How Many Ecosystem Services Do We Actually 'Use'?

Defining the level of "actual use" is relatively straightforward for most provisioning services, as the use itself is extractive, material and, in many cases, it is even already quantified by national statistics/accountings [22,24,79]. However, some variation can be found when looking closer at the definitions, specifically in accounting for the production boundary (or the missing of this), i.e., accounting for the amount of human input in the final ES [80,85]. This is especially problematic when human inputs exceed those received from nature but are not accounted for. Here we might gain the (superficial) impression that ecosystems transformed more heavily by humans are more productive (see detailed discussion in [86]). For example, to take the actually produced amount of grains as an indicator of crop production for this level is possible, but defining that amount that is solely based on nature, is neither easy nor common. In MAES-HU, these aspects were only accounted for "verbally", while the modelling quantified the anthropogenically enhanced crop production.

For some regulating services, defining cascade level 3 is rather challenging [79,87]. It often overlaps with cascade level 2, and it is rather difficult to even express what should be measured if we want to quantify that amount of the potentially available service that is actually used—in a non-extractive way, but deriving some benefit from it in one way or another. For example, for carbon sequestration it is hard to find any amount of $CO_2$ that is sequestered and not useful to us humans (enhancing our well-being) under present day conditions. Thus, here, the complete amount that can be sequestered—for the given ecosystem type, land-use and climatic conditions—could be seen as the actual use. Another option is to account for all carbon additions resulting from land use (e.g., deforestation) and calculate a net carbon sequestration as implemented in MAES-HU as in the IPCC methodology. Other issues occur with flood regulation via water retention in the landscape: one can model the amount of water retained by the ecosystems (i.e., the vegetation and the soil), but for quantifying the amount that enhances human well-being (in terms of the perceived benefit of not being flooded; see [88]), we would need to delineate the part of the retained water that would otherwise cause flooding. Therefore, we chose to model only level 2, while calculations for the complete amount of retained water were

made for a case study area and for urban areas (substituting "actual use"). For other regulating ES, like the filtering of pollutants, the actual-use level strongly depends on the pressure the system is exposed to (i.e., for filtering: the concentration of pollutants) but also on how saturated the system is (i.e., how much further pollutants can be absorbed within the physiological and physical limits of the system). This requires further data on the spatial distribution of pollutants, which were only available for a test site watershed at the required detail for MAES-HU.

In solving the problem of how to assess actual use, a major work-around is to look at how severely people are affected—which is in fact closer related to cascade level 4 (human well-being)—or how many people or what area could benefit from the delivery of this service—which shows rather the demand for that ES [85,87]. These substitutes to actual use are often used in order to represent the importance of regulating services to humans at different locations [85]. A similar approach was applied in MAES-HU for flood regulation in floodplains and drought mitigation, where the potentially affected area was presented at this cascade level.

### 3.3.4. How Can We Link Specific Ecosystem Services with Human Well-Being?

Although most conceptual frameworks include connections towards "benefits", often depicted as contributing to components of human well-being (see e.g., [1,7]), most national ES assessments either do not cover them at all (e.g., [22]) or focus only on some specific components, e.g., material welfare through economic valuation (e.g., [89]), health or social shared values (e.g., [13]). In some assessments, no distinction is made between the different components of well-being; instead, benefits or values of each ES are captured as one overall item (e.g., [14,90]).

None of these studies give a systematic overview of possible linkages between single ES and specific components of human well-being. The connection between the two sides (ES and human well-being) is shaped by certain stakeholder groups (have an impact on the delivery of ES or enjoy the ES), the structured identification of which was attempted in MAES-HU. The results of this exercise can help to communicate the usefulness of ES to the specific stakeholder groups, and more generally to citizens, reflecting on their needs. This expert-based qualitative assessment is the first step towards assessing linkages between ES and components of well-being, a connection that needs to be further illuminated [71,74,91]. An extensive literature review and further empirical research are needed to support these linkages.

### 3.4. Scenarios—How to Plan the Future?

Scenario building enabled us to include all those far-reaching aspects that were excluded from the cascade-level assessments, for example, based on the targeted criteria for ES capacity. In such an exercise, we could assume changes in the basic ET (e.g., reforestations), changes to hydrological connections (e.g., re-flooding of former floodplains) or restoring ecosystem condition in certain areas—changes that we did not want to tackle at the level of ES capacity for keeping the assessment more consistent. In a spatially explicit mapping, this corresponds to what we suggest terming "site capacity" (see Figure 3), thus linking these two strands of present day and future evaluations of landscape's capacity to provide ES. Quantifying not only ET changes derived from the narratives (as in [92,93]), but also changes in their related ES values, opens up more possibilities to reflect on future changes.

Scenario building and evaluation is an interdisciplinary process based on social science methods, and is in fact a decision-supporting tool: it can be used to anticipate social and environmental changes that will likely affect human well-being in the future [13,94]. While it is not strictly part of the MAES process, it complements the assessment in a rather useful and constructive way and is also suggested as part of the Integrated Ecosystem Assessment Framework [32,33]. Scenarios were considered in some other national-level ecosystem services assessments as well (e.g., Spain, Portugal and UK, see [12]). The

method can be used in a wide range of policy contexts, so the range and scale of influencing factors and sectors involved can be narrowed and focused accordingly and applied from regional to continental or global scales (e.g., [24,55]).

*3.5. Integrating and Synthesising Knowledge*

A synthesis towards the end of a MAES process is a major integrating step that was also part of the Hungarian national MAES. This step aimed at shedding light on the interactions between the assessed ES and on the capacities lying in the landscape as a final step after the evaluation of the selected set of ES. This is essential for advancing towards an all-encompassing picture that has the potential to provide balanced guidance on land use and conservation. Synthesizing steps and a similar hotspot analysis as that used in MAES-HU were also performed by some other member states at the national level, e.g., in Greece with a 10 × 10 km resolution [19] or in Germany [59].

For a coherent analysis, clear, comprehensive and practice-oriented definitions of the cascade levels are essential [25,28]. In order to detect potentially conflicting uses of ES, trade-offs and synergies resulting from certain use patterns, it is the actual use level that needs to be analyzed [35]. Several ES form synergies as long as it is the ES provisioning capacity that we regard: for example, timber provisioning and protection from erosion—if the first is actually extracted, the second cannot be provided anymore (see also [35]). The presented considerations regarding ES capacity and the actual use level can help when a set of ES needs to be overviewed, integrated or aggregated.

## 4. Conclusions

Ongoing discussions regarding the ES framework are generally seen as hindering implementation, acceptance and policy uptake [24,27,87]. Working with the ES cascade as a guideline, along which steps of the assessment procedure were drafted, confronted us with conceptual questions that needed to be solved in order to set up a consistent national MAES. Discussing these questions resulted in a set of solutions and insights that are potentially useful for broader application within the ES framework in general and the ES cascade especially.

The presented national mapping and assessment was initiated in compliance with the EU Biodiversity Strategy 2020 target and MAES standards; therefore, the taken approach, its solutions and conceptual suggestions are relevant to other member states within the EU and any countries planning to assess their ecosystem services. The provided results of the mappings can also feed into Natural Capital Assessments complying with SEEA-EA standards.

In the long run, national mappings and assessments of ecosystem services provide a basis for evaluating different land-use options in order to find those solutions that are most useful to society at large at a sustainable level and have the potential to support reaching the United Nations' Sustainable Development Goals.

**Author Contributions:** Conceptualization, Á.V., E.T., E.T.K., B.C. and A.K.-H.; methodology, Á.V., E.T., E.T.K., Á.K., I.A., B.C., E.C., M.K., V.F., L.K.F., P.K., R.L., L.P., R.P., R.R. K.B., A.Z., Z.S. and A.K.-H.; investigation, all authors; resources, M.B., R.L., L.P. and R.P.; writing—original draft preparation, Á.V., E.T., E.T.K., Á.K. and I.A.; writing—review and editing, Á.V., E.T., E.T.K., Á.K., I.A., B.C., V.F., KT, Z.Z. and A.K.-H.; visualization, Á.V.; project administration, L.K.F., K.T., Z.Z. and A.K.-H.; funding acquisition, B.C., L.K.F. and K.T. All authors have read and agreed to the published version of the manuscript.

**Funding:** The project "Strategic investigations on the long-term preservation and development of natural heritage of community importance and on the implementation of the EU Biodiversity Strategy 2020 objective" was funded by the European Regional Developmental Funds as part of the Széchenyi 2020, the Environmental and Energy Efficiency Operative Program and the Competitive Central Hungary Operative Program, grant number KEHOP-4.3.0-VEKOP-15-2016-00001. Funding was also received from the Hungarian National Research, Development and Innovation Office (NKFIH OTKA Grant K 128606).

**Institutional Review Board Statement:** Not applicable.

**Informed Consent Statement:** Informed consent was obtained from all subjects involved in the study.

**Data Availability Statement:** Not applicable.

**Acknowledgments:** We are grateful to all the people who took part in the project; gave their knowledge and opinions in interviews; helped as experts in assessments, to members of the Technical Working Groups and of the Advisory Board, or in managing the project. The project was implemented with the leadership of the Nature Conservation Department of the Agricultural Ministry of Hungary, the Centre for Ecological Research (CER), the Lechner Knowledge Centre, the Institute for Soil Sciences, Centre for Agricultural Research, the Research Institute of Agricultural Economics.

**Conflicts of Interest:** The authors declare no conflict of interest.

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
