# Peer review of "National Ecosystem Services Assessment in Hungary: Framework, Process and Conceptual Questions"

_sustainability, doi:10.3390/su141912847_

Round 1

Reviewer 1 Report

The paper described the process of the national mapping and assessment of ES in Hungary (MAES-HU) from ecosystem type mapping, the selection of relevant ES and their indicators at the cascade levels to their mapping, and then discussd the conceptual considerations in MAES-HU. The manuscript is well organized, the structure is strong and easy to follow, and the process of the national mapping and assessment of ES may be of international relevance.

Good luck!

Author Response

Thank you very much for your appreciation!

Reviewer 2 Report

There are numerous abbreviations in this paper, do ensure that their full spellings are provided, e.g., Line 25, EU is what, European Union?

The contents about the methodology in the Abstract section are really rough, since the whole paper is essentially about the methodology and process of a particular project, the whole paper should be better organized to focus on this topic, and it should be made more reader-friendly.

Some tables are too wide, change the page setup to improve the layout of this paper. In Figure 2, the two angled arrows on the left side would be better replaced with curved ones.

The Conclusions section is not really like a conclusion, and it is more like a summary of the Discussion section.

How reliable or feasible is the framework, process of the national ecosystem services assessment in Hungary? This is a key point of this paper, but there is no available relevant information in the current version.

Reviewer 3 Report

Review Report for: National ecosystem services assessment in Hungary – 2 framework, process and conceptual questions

Thank you for the opportunity to review this paper. In general, it presents an important and interesting framework on how to assess ecosystem services. The authors review methods regarding ecosystem services and propose improvements that are relevant to the Hungarian context. Overall, this paper has the potential to provide an important contribution to the literature; however, it must first be restructured to better fit as a review / methods paper.

In general, the authors should add additional content in the introduction regarding the goals of the work and restructure the discussion section to better link their review of the literature with their own methods in order to communicate strengths and weaknesses. In general, more analysis is needed with respect to how the proposed method compares with the literature.

Additional comments are provided below:

Minor suggestion to change the dash “-“  in the title to a colon “:”.

Abstract

The terms “Strategy to 2020” and “Strategy to 2030” are a bit vague. I would briefly define what these are. Maybe something like “…Strategy to 2030, which are EU plans for …”

“ES Cascade” is a term that might not be familiar to all readers. I suggest replacing this with something more general, even if it’s the case that you define it in the manuscript.

I like that the last line of the abstract discusses the manuscript’s contribution

Introduction

Line 45: Change “in” to “into”

Line 46: You are referencing specific components of the MA without proper background for readers. I suggest adding additional content that sets up the structure of the MA so that readers can more easily follow along.

Line 48: I suggest changing “conditions of ecosystems” to “ecosystem conditions’ to match the shorthand you’re using (EC).

Line 51: Use consistent capitalization for “member states”. In a previous line they are capitalized.

Line 61: use member states here instead of just “states”.

Line 83: Spell out this acronym “SEEA-EA”.

Line 94: before the contributions of the paper are listed, another paragraph is needed to specifically discuss the Hungarian context apart from the overall European one. This will help readers identify the similarities and differences between other efforts in Europe and Hungary, which I think would add to the contribution of this paper.

Methods

 Line 104 – please include a citation for this method

Line 111 – This description is a bit vague, I would recommend revising this sentence for clarity.

Line 112 – What kind of economic valuation? I would include additional content about the methods behind the economic valuation, maybe even an additional paragraph. This could potentially be a very interesting and important part of the paper.

Line 140 – The inclusion of expert opinion in the design of the methods is great. In reality I think this is something more studies should use. I would love to see this more formally represented, maybe as in a table. This is just a suggestion, and you don’t have to respond to this if you can’t find a way, but I think a more formal representation of how expert opinions are included (e.g. table) would be something future studies would look to in how they present their results.

Line 149 – This sentence is a little confusing, I suggest rewording. I like that you have included a citation to the material.

Line 176 – Change “conditions” to “context”.

Line 180 – please find a way to incorporate this list into the paragraph more smoothly.

Lines 278 and 291 – Please find a way to incorporate these lists into the paragraph.

Discussion

Line 384 – This is an interesting line and needs to be expanded on. Why wasn’t the public included, and how does that impact the results of this paper? How does this compare to other surveys?

Line 475 – Please incorporate this list into the paragraph more smoothly.

In general, this section lacks discussions of results. Though there is a lot of content in this paper, and it provides a considerable amount of important insight, this paper probably needs to be reframed in a major way. I have two suggestions, depending on what the authors wish the paper to do:

1.       If this paper is to present results of an analysis, it needs to discuss the data and the results. I do not think this is the case, as the discussion section does not feature any data.

2.       If instead it is a literature review, The introduction needs to be rewritten to frame it in that manner.

3.       If instead it is a methods paper, the paper needs to be reframed accordingly, and in general the discussion section should be a literature review (e.g. 2) should be reviewed with the methods described, and examples of how results differ should be given.

In general, I think it is okay to have this paper be a bit more review oriented, especially if you are proposing a new method, but the introduction must be changed regardless, and the discussion section must be updated to include a better set of comparisons.

 I am also okay if the authors decide to restructure the paper in their own way, given that it provides a more precise statement of the research questions or problems the paper is addressing, and these are more systematically and clearly addressed in the discussion section.

Conclusion

609 – Please expand conclusion to include at least two more paragraphs,

Round 2

Reviewer 2 Report

There are still a number of format errores, e.g.,  [La 507 Notte 2021] in Line 507, [Nedkov 2018] in Line 566.

Author Response

Dear Reviewer,

Thank you for looking thoroughly at the manuscript. As it seems that our referencing software can’t cope with track change we upload a track changed version with incorrect format of citations (as the main docx-file) and also one with all-accepted changes with the right citation formattin as pdf, and for further editing as a “supplementary files” (as there was no other option where this could be provided).

Best regards,

Agnes Vari

(corresponding author)